# Efficacy of a Combination Therapy with Laronidase and Genistein in Treating Mucopolysaccharidosis Type I in a Mouse Model

**DOI:** 10.3390/ijms25042371

**Published:** 2024-02-17

**Authors:** Marcelina Malinowska, Wioletta Nowicka, Anna Kloska, Grzegorz Węgrzyn, Joanna Jakóbkiewicz-Banecka

**Affiliations:** 1Department of Medical Biology and Genetics, Faculty of Biology, University of Gdańsk, Wita Stwosza 59, 80-308 Gdańsk, Poland; marcelina.malinowska@ug.edu.pl; 2InviMed Ferility Clinics, 10 Lutego 16, 81-364 Gdynia, Poland; wiolettak.nowicka@gmail.com; 3Department of Molecular Biology, Faculty of Biology, University of Gdańsk, Wita Stwosza 59, 80-308 Gdańsk, Poland; grzegorz.wegrzyn@ug.edu.pl

**Keywords:** mucopolysaccharidosis type I, glycosaminoglycans, laronidase, genistein, combination therapy

## Abstract

Mucopolysaccharidosis type I (MPS I) is a lysosomal storage disorder caused by α-L-iduronidase deficiency. The standard treatment, enzyme replacement therapy with laronidase, has limited effectiveness in treating neurological symptoms due to poor blood–brain barrier penetration. An alternative is substrate reduction therapy using molecules, such as genistein, which crosses this barrier. This study evaluated the effectiveness of a combination of laronidase and genistein in a mouse model of MPS I. Over 12 weeks, MPS I and wild-type mice received laronidase, genistein, or both. Glycosaminoglycan (GAG) storage in visceral organs and the brain, its excretion in urine, and the serum level of the heparin cofactor II–thrombin (HCII-T) complex, along with behavior, were assessed. The combination therapy resulted in reduced GAG storage in the heart and liver, whereas genistein alone reduced the brain GAG storage. Laronidase and combination therapy decreased liver and spleen weights and significantly reduced GAG excretion in the urine. However, this therapy negated some laronidase benefits in the HCII-T levels. Importantly, the combination therapy improved the behavior of female mice with MPS I. These findings offer valuable insights for future research to optimize MPS I treatments.

## 1. Introduction

Mucopolysaccharidosis type I (MPS I) is a rare lysosomal storage disorder resulting from a deficiency in α-L-iduronidase (IDUA) (E.C.3.2.1.76) activity. This enzyme plays a crucial role in the degradation of glycosaminoglycans (GAGs), specifically dermatan sulfate and heparan sulfate. Insufficient IDUA activity leads to impaired GAG degradation, causing the accumulation of these compounds within lysosomes and resulting in multisystem dysfunction. The spectrum of phenotypic expression, ranging from mild to severe, categorizes affected individuals as having attenuated or severe MPS I, and this distinction influences therapeutic options [1,2].

Recent advances in the biology and function of lysosomes have helped better understand the pathophysiology of mucopolysaccharidoses. The concept that MPS symptoms are a direct consequence of lysosomal swelling due to undegraded GAG has been challenged by new insights into the biological role of GAG and a novel perspective on lysosomes as signaling hubs involved in multiple critical cellular functions. The pathophysiology of MPS I is now perceived to be the result of a complex cascade of secondary events leading to the dysfunction of several cellular processes and pathways, such as an altered membrane composition and its impact on vesicle fusion and transport, secondary substrate storage, impaired autophagy, compromised mitochondrial function and oxidative stress, and the dysregulation of signaling pathways. Characterizing this cascade of secondary cellular events is crucial for a better understanding of the pathophysiology of the disease [3].

One of the first proposed therapies for MPS I was hematopoietic stem cell transplantation (HSCT), which was first performed in 1980 [4]. Currently, the term HSCT includes the use of umbilical cord blood cells or bone marrow as grafts, with HSCT considered the gold standard for severe MPS I cases, ideally undertaken before the age of 2.5 years [5]. Intravenous enzyme replacement therapy (ERT) with recombinant α-L-iduronidase (laronidase) [6,7] is the standard treatment for attenuated disease. Furthermore, it can be used both before and during HSCT. Other therapeutic approaches such as gene therapy or editing, codon stop read-through, and therapies with small molecules are under development [1,8].

A promising therapeutic strategy is the use of small molecules that not only modulate the processes of GAG synthesis or degradation but also cross the blood–brain barrier. Flavonoids are non-specific GAG metabolism modulators that can be used as active compounds in substrate reduction therapies (SRTs). Genistein and kaempferol have been shown to efficiently inhibit GAG synthesis in MPS I, II, IIIA, and IIIB cells [9,10] by modulating the gene expressions of enzymes involved in GAG synthesis and degradation [11,12]. Moreover, both flavonoids stimulate lysosomal biogenesis [11]. Studies have shown that genistein, when used alone or in combination with other flavonoids, inhibits GAG synthesis and reduces storage [13]. However, the efficacy of high-dose genistein aglycone in MPS I is uncertain, as it did not lead to clinically meaningful reductions in biomarkers or improvements in neuropsychological outcomes [14]. Genistein has also been found to partially correct cell cycle disturbances in MPS cells [15,16], suggesting a potential mechanism of action. However, SRT with genistein has shown variable effects on glycan and glycosaminoglycan levels, with changes observed in a cell-type-dependent manner [17]. Although genistein has shown promise in reducing GAG synthesis and improving cell cycle disturbances in MPS, its clinical efficacy and effects on specific disease outcomes require further investigation.

Despite significant advances in the development of treatments, none of them are effective for all symptoms of MPS I [8]. The blood–brain barrier makes the central nervous system (CNS) one of the most difficult systems to treat. ERT and HSCT have limitations and little to no effect on the skeleton, joint, hearing, vision, or neurocognitive decline [18]. In addition, HSCT or gene therapy targeting the CNS only partially corrects the pathology [19]. Therefore, a tempting approach seems to be to combine several treatment strategies simultaneously to obtain the best therapeutic effect on affected tissues. For example, the combination of ERT and HSCT results in improved treatment efficacy in MPS I and MPS VII mice [20,21,22]. The use of both therapies favorably affected the distribution of the enzyme, which reduced the accumulation of GAG even in difficult-to-treat tissues, such as the bone and brain. A short-term administration of laronidase in combination with HSCT is used in severe cases of MPS I, improving patient survival, transplant tolerance, and reducing disease-related complications [23,24,25].

As the blood–brain barrier is the cause of an insufficient delivery of enzymes to the central nervous system, it has been proposed that flavonoids used in SRT more readily cross the blood–brain barrier and correct the neuropathology in MPS I, which cannot be corrected with ERT [26]. Therefore, exploring a combination therapy based on enzyme administration and strategies to reduce excessive GAG influx into lysosomes represents an area in need of investigation.

This study aimed to evaluate the efficacy of a combination therapy involving recombinant α-L-iduronidase (laronidase) and genistein as a treatment for mucopolysaccharidosis type I. Using a mouse model of the disease, we investigated the effects of long-term treatment with laronidase at a dose of 100 U/kg/week, genistein at a dose of 160 mg/kg/day, and a combination of both on GAG storage in visceral organs and the brain, as well as their urinary excretion. Additionally, we assessed the impact of the tested treatments on the serum levels of the heparin cofactor II–thrombin (HCII-T) complex. Finally, we assessed the effect of the combination therapy on CNS physiology by assessing animal behavior in an open field test.

## 2. Results

### 2.1. The Combination of Laronidase and Genistein Reduces Tissue GAG Storage in MPS I Mice

In MPS I, the accumulation of GAG affects various organs, such as the liver, spleen, kidneys, heart, and brain. To evaluate phenotypic and sex-related differences in GAG storage and to assess the effectiveness of treatment, GAG content was quantified in organs collected from 20-week-old MPS I and wild-type mice of both sexes, either untreated or treated with 100 U/kg/week laronidase, 160 mg/kg/day genistein, or a combination of these treatments for 12 weeks.

GAG storage was significantly higher in the livers and spleens of both male and female MPS I mice than in their WT littermates (Figure 1). Interestingly, GAG storage was almost twice as high in the liver and spleen of male MPS I mice than in females of the same age. This difference was not observed in the wild-type animals. Owing to significant sex-related differences in GAG tissue storage, the animals in all subsequent experiments were grouped according to their phenotype (MPS I and WT) and sex (male and female).

To evaluate the effect of laronidase, genistein, and their combination on GAG storage, we quantified the GAG content in tissue lysates of the liver, spleen, kidneys, heart (muscle and valves separately), and brains of MPS I mice and normalized them to the GAG content quantified in the tissues of the untreated MPS I control mice.

All tested treatments reduced GAG storage in the visceral organs of MPS I mice; however, each treatment affected storage in individual organs differently (Figure 2). The response to treatment also varied between sexes.

Laronidase significantly reduced the GAG storage in the liver, spleen, and kidneys of male and female MPS I mice compared with that in the untreated MPS I littermates (Figure 2). The highest, a 3–5-fold, reduction in GAG content was observed in the liver and spleen. The heart showed large individual variations in response to laronidase; the GAG amount decreased in some animals and increased in others.

Genistein significantly decreased GAG storage in the liver of MPS I mice of both sexes, and in the spleen of male MPS I mice compared to that in untreated MPS I mice (Figure 2). However, this treatment was significantly less effective than the laronidase or combination therapy. In the kidneys, GAG storage remained unchanged in males and increased in females after genistein treatment. This increase was significant compared with that of laronidase and combination therapy.

Combination therapy significantly reduced GAG storage in the liver, spleen, and kidneys of male and female MPS I animals (Figure 2). The effectiveness of this treatment was comparable to that of laronidase monotherapy in the liver and spleen but was less effective in female kidneys. Compared to both monotherapies, combination therapy significantly reduced GAG storage in the heart valves of MPS I females and the heart muscle of MPS I males compared to those of their untreated MPS I littermates. In males, this therapy was significantly more effective than genistein and laronidase monotherapies. In females, this therapy tended to be the most effective compared to the other therapies, but statistical significance was not achieved.

The effects of the tested treatments were also observed in wild-type animals of both sexes (Appendix A). In males, the combination therapy resulted in a significant increase in GAG levels in the spleen, heart muscle, and brain, whereas genistein monotherapy reduced GAG levels in the kidneys (Appendix A). In females, genistein, alone or in combination with laronidase, resulted in a significant reduction in the GAG levels in the spleen. Moreover, combination therapy increased GAG levels in the brains of wild-type female mice (Appendix A).

The tested treatments differentially affected GAG storage in the brains of MPS I mice, and the response was sex specific (Figure 2). Laronidase was ineffective in reducing GAG storage in the brains of MPS I mice of both sexes compared with untreated MPS I mice. Genistein significantly reduced brain GAG storage, but only in female MPS I mice. In response to the combination treatment, the brains of both male and female MPS I mice showed large individual variations. In some animals, GAG storage decreased after treatment, whereas in others, it increased, however, no statistically significant differences were observed between the treated and untreated MPS I animals. Interestingly, genistein monotherapy in females was significantly more effective than in combination therapy.

Combination therapy was as effective as laronidase monotherapy in the liver, spleen, and kidneys, suggesting that the therapeutic effect might be related to laronidase activity. The addition of genistein to laronidase resulted in an increased effectiveness in reducing GAG storage in the heart valves or muscles, but the effect was sex dependent. Surprisingly, the combination therapy decreased the effectiveness of genistein in reducing GAG storage in the brains of female mice.

These results indicate that each treatment reduced GAG storage, but had distinct effects on individual organs, and the response was sex dependent.

### 2.2. Combination of Laronidase and Genistein Reduces Hepatosplenomegaly in MPS I Mice

Increased GAG accumulation in the liver and spleen cells leads to the enlargement of these organs, and this symptom is classified as hepatosplenomegaly in patients with MPS I. To evaluate this symptom in MPS I mice and to assess the effectiveness of the tested treatments, both organs were weighed after 12 weeks of treatment.

MPS I mice were characterized by increased liver and spleen weights compared with their wild-type littermates (Appendix A). The livers of male mice were approximately 20% heavier than those of female mice, whereas females had 20–30% heavier spleens than males.

Both laronidase and combination therapy significantly reduced the liver weight of MPS I mice of both sexes and spleen weight of MPS I females compared with those of untreated control animals (Figure 3). Genistein did not affect the organ weight. The addition of genistein to laronidase reduced enzyme effectiveness in the male liver.

These results indicate that hepatosplenomegaly can be reduced through both laronidase and combination therapy, but the effectiveness did not increase with the addition of genistein.

### 2.3. The Combination of Laronidase and Genistein Reduces Urinary GAG Excretion in MPS I Mice

Most patients with MPS I excrete excess GAGs in urine or exhibit different relative proportions of GAG types compared to healthy individuals. To evaluate urinary GAG excretion in the murine model, GAG levels were quantified in urine samples collected from MPS I and wild-type mice of both sexes and standardized to the level of creatinine, which provides a constant value based on the body weight, muscle mass, and sex of the individual animal.

GAG excretion was assessed in the urine of 20-week-old mice to evaluate phenotypic and sex-related differences. Both male and female MPS I mice exhibited increased urinary GAG excretion compared with their wild-type littermates (Appendix A). The average urinary GAG excretion in female mice with MPS I was approximately 20% higher than that in male mice with MPS I.

To evaluate the effectiveness of the tested treatments, urinary GAG excretion was assessed in MPS I mice before treatment, after 6 weeks of treatment, and after 12 weeks of treatment (8-, 14-, and 20-week-old mice, respectively). Each treatment had its own procedural control because we assumed that the volume of administered fluids might have an impact on the amount of urine and, therefore, the level of GAG excretion.

The level of GAG in the urine remained low and constant in wild-type mice but increased with age in MPS I mice (Figure 4). Laronidase administration significantly reduced urinary GAG excretion compared with that in the wild-type littermates. However, the GAG excretion remained unchanged after genistein treatment. Combination therapy reduced the urinary GAG excretion regardless of the sex of the MPS I mice; however, the relative decrease in GAG excretion between treated and untreated MPS I mice was lower in older animals than that obtained with laronidase monotherapy.

### 2.4. Laronidase Reduces the Serum Levels of Heparin Cofactor II–Thrombin Complex in MPS I Mice

The heparin cofactor II–thrombin (HCII-T) complex plays a regulatory role in blood clotting. An elevated HCII-T level in patients with MPS serves as a valuable biomarker for disease diagnosis and monitoring. To evaluate the relationship between this biomarker and disease phenotype and progression in MPS I mice, and to assess the effects of the tested treatments, we measured the levels of HCII-T in the serum obtained from 8- and 20-week-old MPS I and wild-type mice of both sexes, either untreated or treated with laronidase, genistein, or combination therapy for 12 weeks.

In control animals, the level of HCII-T was at least 10 times higher in the MPS I mice of both sexes than in their wild-type littermates (Appendix A). Moreover, its level significantly increased with age in MPS I animals of both sexes, as the disease progressed. In wild-type animals, these levels also slightly increased with age; however, the measured levels were very low and the change over time was not significant.

The observed differences in the serum levels of the HCII-T complex make it a good biomarker for monitoring disease progression in MPS I mice of both sexes over time.

The administration of laronidase for 12 weeks resulted in an approximately two-fold decrease in serum HCII-T complex levels in MPS I mice of both sexes compared to that of the control MPS I animals of the same age that received no treatment (Figure 5). Genistein did not affect the level of the complex. Surprisingly, when both treatments were used simultaneously, the level of HCII-T did not change in MPS I males but increased in MPS I female mice.

Although laronidase was effective in reducing the levels of the HCII-T complex, the addition of genistein in combination therapy abolished or worsened this effect.

### 2.5. Effects of Laronidase, Genistein, and Combination Therapy on the Behavior of MPS I Mice

The accumulation of GAGs in the brain can lead to cognitive impairment and behavioral changes. To evaluate phenotypic and sex-related differences in behavior and assess the effectiveness of the tested treatments, an open field test was performed. During the test, locomotor activity, exploration, anxiety-like behaviors, and the habituation of animals were assessed in MPS I and wild-type mice of both sexes. In a preliminary experiment, the mice were initially assessed at eight weeks of age, with subsequent testing every four weeks until they reached eight months of age. This extended timeframe was employed to pinpoint the critical point at which the behavioral distinctions between MPS I and wild-type animals become pronounced enough to serve as a meaningful reference for monitoring the efficacy of the tested treatments in ameliorating neurobehavioral symptoms associated with MPS I. Male MPS I mice did not show statistically significant differences in behavior, neither in the level of activity nor in the exploration of the central part of the arena (Appendix A, left panel). The behavior of females of the same age showed statistically significant differences between MPS I and wild-type mice (Appendix A, right panel). The difference was first noticeable at the age of four months and was significant for six-month-old animals regarding the frequency of entering and time spent in the center zone, but not in the case of total activity. Moreover, we observed a reversal in the pattern of behavior measured using the indicated parameters around the fourth month of age. Younger MPS I females covered a shorter distance during the test, spent less time in movement, and entered the center of the arena less often; with age, these parameters remained at a relatively stable level. In wild-type females, all examined parameters gradually decreased with age, which was not observed in males.

In the next part of the initial assessment of the behavior of our animal model, we investigated how the animals’ behavior changed while performing a thirty-minute test at five-minute intervals. MPS I mice showed no statistically significant differences in the total distance traveled or time spent moving (Appendix A). However, the animals entered the central part of the arena more often at both the tenth and twentieth minutes (Appendix A), which corresponded to reduced anxiety-like behavior. The increased frequency of entering the central part of the arena and the duration of stay indicate that MPS I females had a skewed natural fear of open spaces, which is recognized as a potential threat.

To evaluate the effects of laronidase, genistein, and combination therapy on neurobehavioral symptoms associated with MPS I, the female mice underwent an open field test at six months of age after receiving four months of treatment. The results are shown in Figure 6.

Compared with untreated mice, laronidase and combination therapy did not affect the locomotor activity of MSP I female mice, as measured using the total distance covered and time spent in motion (Figure 6A,B). In contrast, mice treated with genistein showed reduced locomotor activity according to these measures.

Genistein and combination therapy resulted in a reduced frequency of entries into the central zone compared to untreated MPS I mice (Figure 6C). However, there was no significant difference in the percentage of time spent in the central and border zones of the arena between the different treatments (Figure 6D). Only a trend was observed in animals receiving genistein alone or in combination with laronidase, which spent less time in the center. The activity of MPS I mice in the central zone was comparable to that of untreated animals and did not vary significantly between the treatment groups. However, in the periphery, genistein-treated MPS I mice remained motionless for significantly longer periods than untreated mice and those treated with laronidase or combination therapy (Figure 6E).

None of the treatments affected the locomotor activity of the wild-type female mice (Appendix A). Regardless of the treatment received, these mice entered the center of the arena at similar frequencies and spent a comparable amount of time (Appendix A). Similarly, the proportion of time that the animals moved or remained motionless in both the center and border zones showed no significant differences (Appendix A).

## 3. Discussion

In this study, we evaluated the therapeutic efficacy of combining 100 U/kg/week of laronidase with 160 mg/kg/day of genistein to address the pathology associated with MPS I in a murine model. Our findings indicate that this combination therapy significantly ameliorated various pathological manifestations of MPS I. These include a reduction in GAG storage across multiple organs, decreased hepatosplenomegaly, reduced urinary GAG excretion, and an alleviation of neurobehavioral symptoms following a 12-week treatment period. However, the treatment outcomes were variable, and in certain instances, combination therapy decreased the effectiveness of the monotherapies.

Significantly, monotherapy with laronidase alone resulted in a considerable decrease in GAG storage in the liver, spleen, and kidneys, along with a reduction in liver and spleen weights and a reduction in the GAG excreted in the urine of MPS I mice of both sexes. These results are consistent with those of previous studies [27,28,29]. However, ERT with laronidase did not decrease GAG storage in cardiac tissues, in either the valves or muscle. It is possible that laronidase may contribute to stabilizing cardiac function and does not facilitate the reversal of pathological alterations within the heart. Therefore, early diagnosis and a prompt initiation of ERT are crucial [30,31]. Additionally, despite its efficacy in reducing GAG storage in visceral organs, laronidase did not reduce GAG accumulation in the cerebral tissues of MPS I mice. The clinically recommended dose of laronidase is ineffective at crossing the blood–brain barrier, and evidence suggests that only high doses of the enzyme yield the desired therapeutic effect in MPS I mice [28].

In contrast to laronidase, monotherapy with 160 mg/kg/day genistein effectively reduced GAG storage in some visceral organs, as well as in the brain, and ameliorated behavioral symptoms in female MPS I mice. Genistein, a small-molecule compound, inhibits GAG synthesis [9] and modulates lysosomal metabolism and biogenesis [11,12]. This compound has also been shown to cross the blood–brain barrier [32]; therefore, it has been tested as a therapy based on substrate reduction as an alternative to ERT. Genistein was shown to be effective in reducing GAG storage in the visceral organs of MPS II [33] and MPS IIIB mice [34], and in improving the behavior of MPS III mice [35]; nevertheless, some adverse effects were also observed in MPS I mice [36]. In this study, we confirmed that genistein reduced GAG storage in the livers of both male and female MPS I mice and in the spleens of male MPS I mice. The observed sex-dependent response to genistein treatment could be attributed to variations in epidermal growth factor receptor (EGFR) quantities [37], which play a crucial role in maintaining optimal GAG synthesis rates [38,39]. The binding of EGF to its receptor is directly proportional to the amount of testosterone [40]. Genistein acts as an inhibitor of EGF tyrosine kinase [41], thereby reducing the efficiency of GAG synthesis [42]. Remarkably, at the administered dose, genistein significantly reduced GAG levels in the kidneys of male and spleen of female wild-type mice, although it did not affect the weight of the liver or spleen, or reduce the urinary GAG levels in MPS I mice.

Importantly, genistein reduced GAG levels in the brains of female MPS I mice and affected their behavior. Treated animals exhibited calmer demeanors, less erratic movement, and increased periods of motionlessness within the confines of a relatively safe environment, in contrast to untreated MPS I controls. No significant behavioral changes were observed in wild-type animals. These outcomes suggest that a long-term administration of high-dose genistein may facilitate its entry into the central nervous system, where it potentially slows neurodegeneration by decreasing GAG synthesis. Genistein is a compound with a broad spectrum of activities including anti-inflammatory, antioxidant, and stress-reducing effects [43,44,45]. These properties may contribute to improving the behavior of MPS I mice.

Combination therapy facilitates effective GAG clearance in the liver and spleen; however, this effect is predominantly attributed to laronidase. Importantly, this treatment effectively reduced GAG storage in the heart muscle or valves. Here, the therapeutic properties of genistein are evident. This observation is crucial given that cardiorespiratory issues are the primary cause of mortality in young MPS I patients [46]. The accumulation of undegraded substances in cardiac tissues contributes to cardiomyopathy, endocardial fibrosis, and thickening of the valves and is associated with the release of proinflammatory cytokines and metalloproteinases and the activation of macrophages, ultimately leading to tissue damage [47]. Genistein exerts a protective effect against myocardial damage [48]. Additionally, in over 90% of patients, laronidase induces the production of specific anti-drug IgG antibodies [49], and this immune response can alter enzyme distribution and tissue uptake [50]. Thus, the anti-inflammatory properties of genistein [51] may inhibit enzyme-induced immune responses, leading to better enzyme penetration into tissues.

We used HCII-T serum levels as a biomarker to monitor the effectiveness of the tested treatments. HCII-T levels correlate with disease severity and therapeutic responsiveness in MPS mice and humans [52,53,54]. We confirmed higher levels of HCII-T in MPS I mice than in their wild-type littermates. The formation of the HCII-T complex is upregulated by GAGs, particularly dermatan sulfate, by interfering with antithrombin, an enzyme responsible for the degradation of the HCII-T complex. Surprisingly, laronidase reduced HCII-T levels in MPS I mice of both sexes, whereas genistein had no effect. Laronidase effectively degrades GAGs, lowering their levels. This normalizes the balance between HCII-T formation and degradation in the bloodstream, ultimately reducing its levels in treated mice with MPS I [55]. Genistein primarily inhibits GAG synthesis [9], and its lack of effect on the HCII-T level suggests that merely reducing GAG synthesis may not be sufficient to affect complex formation. This could be because genistein action does not significantly alter existing GAGs that contribute to HCII-T formation.

Unexpectedly, combination therapy did not reduce HCII-T levels in MPS I mice. The effect of laronidase was abolished by genistein, and HCII-T levels remained unchanged in males but increased in females. This difference between males and females is intriguing and suggests a sex-specific response to combination therapy, which could be attributed to several factors, including hormonal influences, genetic differences, or variations in drug metabolism between the sexes. For example, estrogen is known to influence the metabolism and efficacy of various drugs. This could potentially play a role in the differential response of female mice to combination therapy. The increase in the HCII-T levels in female MPS I mice after combination therapy suggests a complex interplay involving laronidase, genistein, and sex-specific metabolic or hormonal pathways. It is possible that in female mice, combination therapy could induce changes in the GAG metabolism or turnover, increasing the substrates or cofactors necessary for HCII-T complex formation.

The indication of sex-related differences in disease progression in the MPS I mouse model is a notable finding of the present study. The GAG levels in the liver, spleen, kidneys, heart, and brain were elevated in MPS I animals compared to those of their wild-type littermates, as previously demonstrated [27,29,56]. However, we found that the extent of GAG storage was sex-dependent, with male mice displaying higher GAG concentrations in the liver and spleen than females. Moreover, the liver was considerably larger in males, whereas the spleen was larger in females. Similarly, male MPS IIIB mice have been shown to store more GAG in the liver, but not in the spleen [34]. This may be associated with the higher number of EGFRs present in the hepatocytes of male rodents [37]. Because EGFR activity is required to maintain the maximum efficiency of GAG synthesis [38,39], its rate may be higher in males, and when combined with impaired degradation, it may result in higher basal liver GAG storage. Sex-related differences in liver and spleen GAG storage may also result from the deposition of different classes of GAGs in various tissues. In MPS III, heparan sulfate accumulates in the brain, liver, and kidneys. Conversely, MPS I is characterized by the storage of both heparan sulfate and dermatan sulfate, with dermatan sulfate primarily stored in the liver and spleen [57]. Thus, the progressive accumulation of GAGs leads to hepatosplenomegaly in MPS I mice.

We found that animal behavior varied according to genotype, age, and sex. Interestingly, mice with MPS I showed stable levels of activity and anxiety-related behaviors over time, in contrast to wild-type animals, for which both parameters progressively declined with age. A similar decrease in activity in older wild-type mice was observed in other studies [58]. In our behavioral tests, younger MPS I female mice displayed reduced activity and diminished the exploration of the open center zone, aligning with the behaviors characteristic of MPS I progression [59]. Starting at four months of age, female MPS I mice showed increased locomotor activity compared to their age-matched wild-type littermates; however, male MPS I mice did not show such a difference. However, these results contradict other studies reporting no significant behavioral differences based on sex [60]. These discrepancies may be attributable to variations in the mouse models of the disease and the methodologies of the open field test. Currently, numerous mouse models are available for preclinical studies of potential therapeutics for neurodegenerative lysosomal storage diseases, including four models of MPS I [61]. In our study, the knockout model was utilized, and the open field test was conducted for over 30 min using an automated recording system. For example, two studies [59,62] also used *Idua*^−/−^ mice, but the arena in which the animals were located was divided into sectors differently, and the percentage of crossings in the center squares compared to the total crossings and the number of fecal pellet tests for anxiety were manually counted. The test was conducted for five minutes. Another study utilized the same system in their open field test but limited observations to ten minutes and used the *Idua*-W392X mouse model [63]. These differences may account for the observed discrepancies between the various studies. Considering these variables, we decided to analyze all results as relative values or percentages normalized to an internal control for each study group to ensure comparability.

The strength of our study lies in the methodological approach, wherein subjects were classified according to genotype (MPS I vs. wild-type) and sex, recognizing potential differences in treatment responses. This classification underscores the critical role of sex-related physiological differences in modulating the responses to pharmacological interventions. Therefore, the identification of sex-specific differences in disease progression and treatment response in our study provides valuable insights, highlighting the need for personalized therapeutic approaches for MPS I.

Monotherapy with either laronidase or genistein showed significant reductions in the GAG accumulation in different organs, but each had its limitations, such as the inability of laronidase to decrease GAG levels in cardiac tissues and the brain and the sex-dependent response to genistein treatment. Combination therapy enhanced GAG clearance in organs, including the heart, where the therapeutic properties of genistein were particularly noticeable. However, this combination did not reduce HCII-T levels in MPS I mice, and in some cases, these levels increased, suggesting complex interactions that may involve sex-specific metabolic or hormonal pathways.

Our study highlights the potential for combination therapies in the treatment of MPS I; however, the varied effectiveness in different tissues underscores the need for a detailed analysis of drug–drug interactions, for example, via isobolographic analysis [13], to optimize treatment strategies. Recent advancements in MPS I therapeutic approaches have highlighted the potential of pharmacological chaperones and reversible inhibitors to enhance treatment efficacy. For instance, small-molecule protein stabilizers significantly reduce heparan sulfate accumulation in MPS I cells by stabilizing exogenous α-L-iduronidase [64]. Furthermore, the use of pharmacological chaperones, in combination with reversible inhibitors of α-L-iduronidase, has been substantiated by their success in other lysosomal storage diseases. In Fabry disease, the use of pharmacological chaperones enhances α-galactosidase A activity, leading to improved clinical outcomes. Similarly, in Gaucher disease, the use of these chaperones to stabilize glucocerebrosidase has significantly contributed to the efficacy of treatment protocols. Iacobucci et al. showed that such combination therapies could significantly improve therapeutic outcomes, offering a new avenue for treatment strategies for lysosomal storage disorders [65]. These findings suggest that incorporating pharmacological chaperones and reversible inhibitors into the treatment regimen for MPS I could offer a synergistic effect, potentially overcoming the limitations of current therapies and paving the way for more effective disease management.

In summary, the observed sex-dependent responses and differential effects of therapies on various tissues and symptoms emphasize the complexity of treating MPS I and the necessity for personalized therapeutic approaches. The absence of a histopathological analysis and the preliminary nature of exploring sex-specific response patterns to treatment are notable limitations, indicating the need for further clinical research to validate these preclinical findings in human MPS I therapy. The extrapolation of these findings from animal models to human clinical contexts requires a cautious interpretation of the inherent biological divergences between species. This emphasizes the urgent need for further clinical research to ascertain the applicability of these preclinical insights to human MPS I therapy.

## 4. Materials and Methods

### 4.1. Mouse Colony Maintenance and Treatment

The MPS I knockout murine model, strain B6.129-*Iduatm1Clk*/J [56], was purchased from Jackson Laboratory (Bar Harbor, ME, USA). The breeding colony was established and maintained from heterozygous mating pairs at the Tri-City Academic Animal Experimental Research and Service Center (Medical University of Gdańsk, Gdańsk, Poland). Animal care and all experimental procedures were conducted according to the guidelines for animal welfare and approved by the Polish Local Ethical Committee protocol code no. 32/2011. Mice were housed under specific pathogen-free (SPF) conditions at a constant temperature of 22 ± 2 °C and 40–50% humidity, in a 12/12 h light/dark cycle, with food and water ad libitum. DNA from ear punches was isolated using the EXTRACTME DNA Tissue Kit (Blirt, Gdańsk, Poland), and the genotype was established via PCR using allele-specific oligonucleotides (ASOs) (Thermo Fisher Scientific, Glasgow, UK) with the following sequences:IMR1451: 5′-GGAACTTTGAGACTTGGAATGAACCAG-3′;IMR1452: 5′-CATTGTAAATAGGGGTATCCTTGAACTC-3′;IMR1453: 5′-GGATTGGGAAGACAATAGCAGGCATGCT-3.

Amplification was performed in a 15 μL reaction volume, comprising 0.2 mM dNTPs (Roche Applied Science, Indianapolis, IN, USA), 1 μM of each primer, 1 mM betaine (Sigma-Aldrich, Darmstadt, Germany), 2 mM MgCl₂, 1× NH₄ reaction buffer, and 0.03 units of BIOTAQ DNA polymerase (Bioline, London, UK), with 1 μL of the DNA template. PCR was performed on an Eppendorf PCR machine (Eppendorf, Hamburg, Germany). The thermal cycle was programmed for 3 min at 94 °C for initial denaturation, followed by 35 cycles of denaturation at 94 °C for 30 s, annealing at 65 °C for 1 min, extension at 72 °C for 1 min, and final elongation at 72 °C for 2 min. The PCR products were examined via electrophoresis at 100 V for 30 min on a 1% (*w*/*v*) agarose gel in 1× TAE buffer. Genotyping was performed based on the observed band patterns: the wild-type (WT) genotype exhibited a band at 500 bp, the MPS I mutant displayed a band at 350 bp, and heterozygous mice showed bands corresponding to both sizes. Wild-type and mutant (MPS I) mice were used for the in vivo experiments.

Eight-week-old male and female MPS I and WT mice were assigned to one of four study groups (*N* = 6 animals per group). All animals received a Ssniff R/M-H Low-phytoestrogen diet containing 34% protein, 57% carbohydrates, and 9% fat without soy additives (Ssniff Spezialdiäten GmbH, Soest, Germany). Depending on the study group, the mice received the following:Genistein (Pharmaceutical Research Institute, Warsaw, Poland) via daily oral gavage at a dose of 160 mg/kg as a suspension in saline (Polpharma SA, Starogard Gdański, Poland);Recombinant human α-L-iduronidase (laronidase) that was purchased as Aldurazyme (BioMarin Pharmaceutical Inc., Novato, CA, USA) via weekly intraperitoneal (IP) injections at a dose of 100 units/kg;Genistein, 160 mg/kg/day and laronidase, 100 units/kg/week;Control animals received saline via IP injection, oral gavage, or both routes of administration.

The administration of the tested treatments was initiated in eight-week-old mice and continued for 12 weeks. Weight was measured weekly, and urine and blood samples were collected at the start and end of the experiment. The behavior of the animals was examined after two and four months of treatment (i.e., four- and six-month-old mice, respectively). At the end of the experiment, the mice were sacrificed, and their organs (brain, liver, spleen, kidneys, and heart) were collected for biochemical analysis. The liver and spleen were weighed at the time of sacrifice.

### 4.2. Tissue and Sample Preparation

The mice were euthanized via cervical vertebral dislocation, and the organs were collected. The brain was divided into two hemispheres, with one hemisphere further sectioned into five parts encompassing the olfactory bulbs, cortex, hippocampus, midbrain, and cerebellum. Three segments were obtained from the different liver lobes. The spleen was divided into three parts: head, stem, and tail. The kidneys and hearts were collected. Before biochemical analysis, the heart was dissected into a portion containing valves and a segment of heart muscle, whereas only the middle part of the kidneys was sampled. All tissues were snap-frozen and stored at −80 °C for biochemical analysis.

Tissue samples were disintegrated in 1 mL Dulbecco’s phosphate-buffered saline (DPBS) (Gibco, Thermo Fisher Scientific, Glasgow, UK) through three cycles of mechanical homogenization at 16,000 rpm for 1 min using the Dual Processing System DPS-20 (PRO Scientific Inc., Oxford, CT, USA). To enhance glycosaminoglycan (GAG) recovery, the spleen, kidney, and heart samples were sonicated thrice at 75% power for 20 s. Additionally, brain tissue was subjected to papain digestion (Sigma-Aldrich, Darmstadt, Germany) by combining the homogenate at a 1:2 ratio with 0.1 mg/mL papain solution (prepared in 0.2 M Na_2_HPO_4_-NaH_2_PO_4_ buffer, pH 6.4, containing 100 mM sodium acetate, 10 mM EDTA, and 5 mM cysteine-HCl) and incubating at 65 °C for 3 h with periodic mixing, followed by enzyme inactivation at 90 °C for 10 min. Despite these variations, debris were removed from the tissue homogenates via centrifugation at 5000× *g* for 10 min, and the resulting clear supernatants were used for GAG quantification. Urine samples were centrifuged at 3000× *g* rpm for 10 min to eliminate debris and subsequently diluted 5- to 50-fold in distilled water (depending on the phenotype, MPS I, or wild-type) before GAG quantification.

### 4.3. Glycosaminoglycan Quantification Assay

All analyses were conducted in triplicate in a blinded manner. The quantification of glycosaminoglycan content involved assessing 100 μL aliquots of tissue homogenates and urine using the Blyscan–sulfated glycosaminoglycan (sGAG) assay kit (Biocolor Ltd., Carrickfergus, UK), following the manufacturer’s protocol. An aqueous solution of 10 mg/mL chondroitin sulfate (Sigma-Aldrich) was used as the reference standard. Absorbance was measured at 660 nm using a Victor3 1420 Multilabel Plate Reader (PerkinElmer, Waltham, MA, USA). Tissue GAG content was determined from the calibration curve (prepared with chondroitin sulfate amounts ranging from 0 to 5.0 μg in increments of 0.5–1.0 μg) and normalized for DNA concentration, measured using the Quanti-iT PicoGreen dsDNA Reagent (Thermo Fisher Scientific, Waltham, MA, USA) for tissue homogenates, or creatinine concentration, measured using the QuantiChrom Creatinine Assay Kit (BioAssay Systems, Hayward, CA, USA) for urine samples.

### 4.4. Heparin Cofactor II–Thrombin (HCII–T) Complex Quantification Assay

Blood samples were obtained from the submandibular vein without anticoagulant agents and incubated at room temperature for 30 min to facilitate clotting, followed by centrifugation at 3000× *g* for 15 min at room temperature. Serum was collected and stored at −80 °C. The heparin cofactor II–thrombin (HCII–T) complex in serum was quantified through a sandwich enzyme-linked immunosorbent assay (ELISA) using the Matched-Pair Antibody Set for ELISA of human Thrombin-Heparin Cofactor II Complex (Affinity Biologicals Inc., Ancaster, ON, Canada) as described previously [52]. Before quantification, sera obtained from wild-type or MPS I animals were diluted 100- and 500-fold, respectively, using the ELISA Neptune Sample Diluent (AbD Serotec, Kidlington, UK). A calibration curve ranging from 5 to 200 pM was prepared by diluting the standard HCII–T complex. All samples were assayed in duplicate.

### 4.5. Open Field Test

The tests were always performed between 9 am and 10 am to ensure constant conditions and minimize the influence of external factors on animal behavior. Mice were individually placed in an illuminated, polycarbonate, opaque arena (Noldus Information Technology, Wageningen, The Netherlands) measuring 27.3 × 27.3 × 20.3 cm, conventionally divided into the border and central zones, and were recorded for 30 min. The recorded data were analyzed using EthoVision XT 11.5 software (Noldus Information Technology, Wageningen, The Netherlands).

### 4.6. Statistical Analysis

Data were expressed as the mean ± standard deviation. The Grubbs test was used to identify single outliers (two-sided). The Shapiro–Wilk test was used to check for a normal distribution. The Student’s *t*-test or one-way ANOVA with Tukey’s HSD post hoc test was used for normally distributed data; in other cases, the Mann–Whitney U test or Kruskal–Wallis test with Dunn’s post hoc test was used for comparisons between groups. Statistical significance was set at *p* < 0.05.

## 5. Conclusions

The results of this study regarding the effectiveness of laronidase and genistein combination therapy for MPS I showed that although this therapy ameliorated pathological symptoms in many organs and improved neurobehavior, the rate of success was not always better than that of either monotherapy. The main improvements in the analyzed parameters were attributed to laronidase, highlighting its key role in combination therapy. Furthermore, our findings revealed sex-specific differences in the responses to treatment, emphasizing the need for further research into personalized pharmacological strategies and therapeutic approaches. These insights provide a foundation for future research aimed at optimizing treatment protocols and improving the quality of life of patients with MPS I.

## Figures and Tables

**Figure 1 ijms-25-02371-f001:**
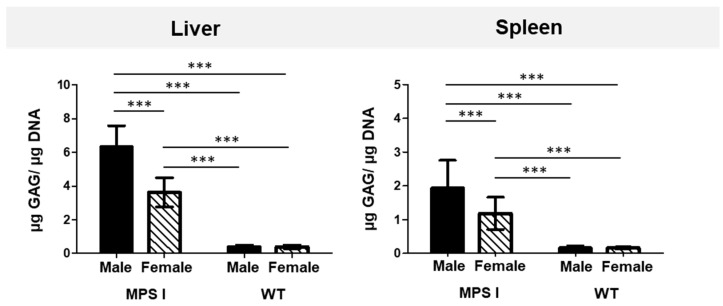
GAG levels in the liver and spleen of 20-week-old MPS I and wild-type (WT) mice of both sexes. Tissue GAG levels were normalized to the amount of DNA. The results are presented as mean ± standard deviation. *N* = 18 animals per group. Significance: *** *p* < 0.001 between sexes and phenotypes (Student’s *t*-test).

**Figure 2 ijms-25-02371-f002:**
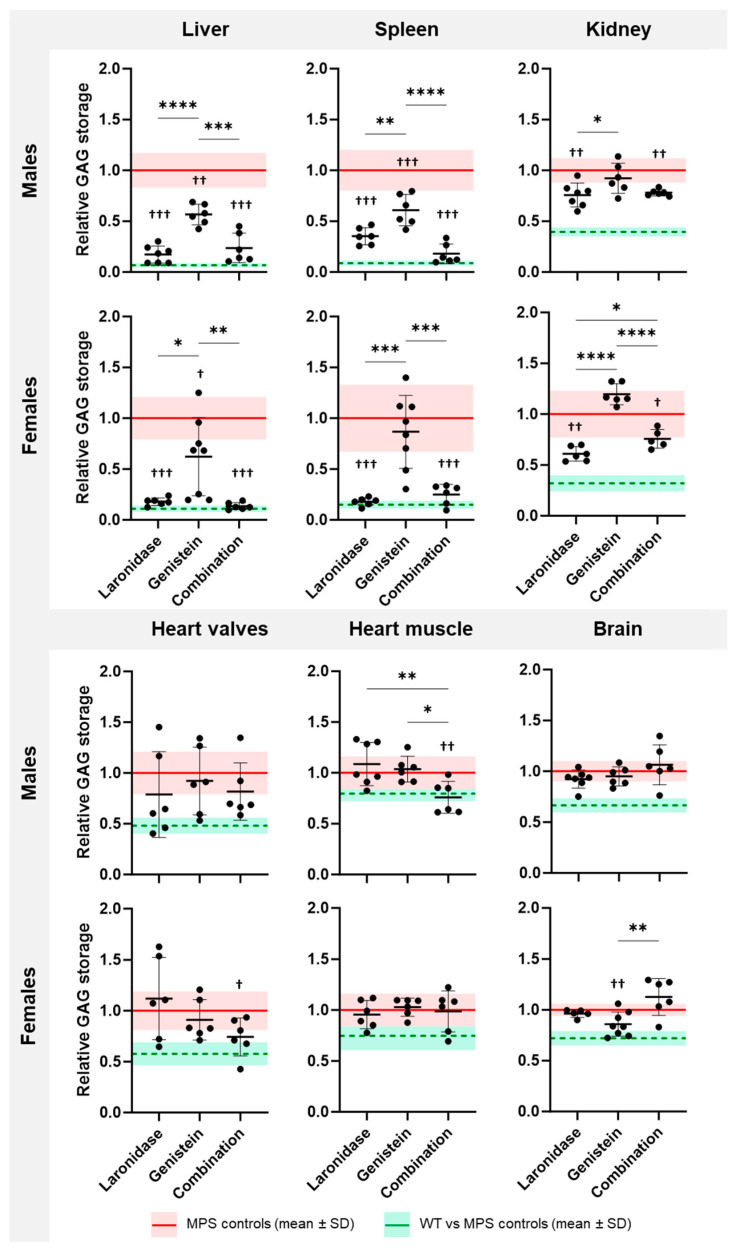
GAG storage in the organs of 20-week-old MPS I mice after treatment with laronidase, genistein, or their combination. GAG levels were determined in the liver, spleen, kidneys, heart valves, heart muscle, and brain obtained from animals of both sexes receiving 100 U/kg/week laronidase, 160 mg/kg/day genistein, or a combination of both for 12 weeks and were normalized to those of untreated MPS I mice. The graphs represent the results obtained for individual animals and show the mean ± standard deviation (SD). GAG storage in untreated MPS I mice is presented as a red horizontal line ± SD. GAG storage in wild-type (WT) animals is presented relative to that in untreated MPS I animals (green horizontal line ± SD). *N* = 6 animals per group. Significance: * *p* < 0.05, ** *p* < 0.01, *** *p* < 0.001, or **** *p* < 0.001 between treatments (Tukey’s post hoc test after ANOVA or Dunn’s post hoc test after Kruskal–Wallis test); ^†^ *p* < 0.05, ^††^ *p* < 0.01, or ^†††^ *p* < 0.001 between treated and untreated MPS I mice (Student’s *t*-test).

**Figure 3 ijms-25-02371-f003:**
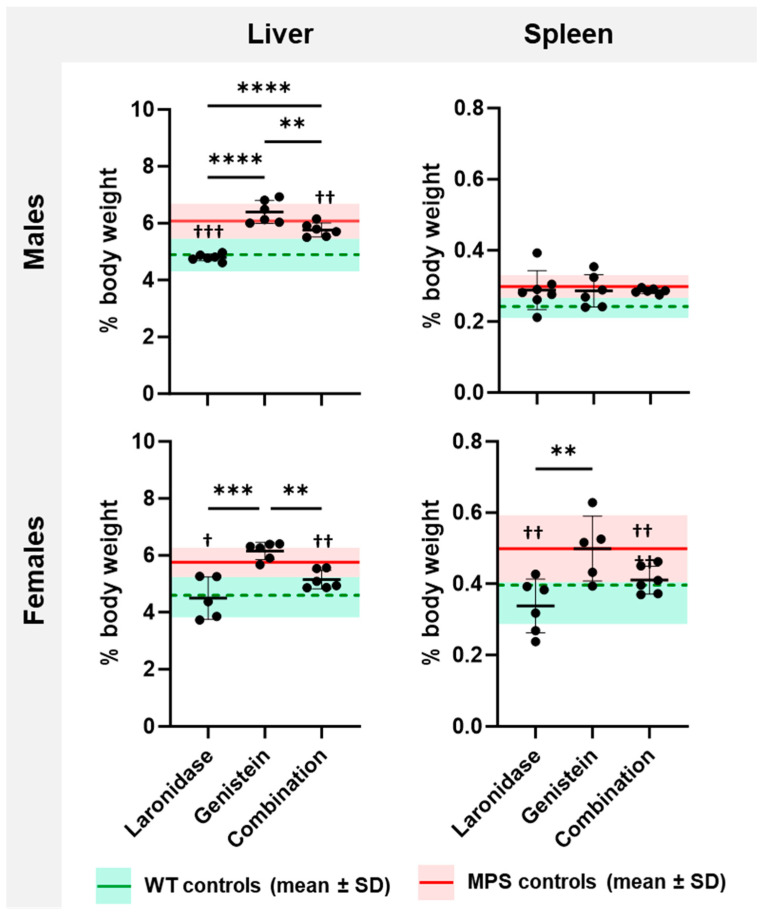
Liver and spleen weights of 20-week-old MPS I mice after treatment with laronidase, genistein, and combination therapy. Organ weights were measured following a 12-week administration of 100 U/kg/week laronidase, 160 mg/kg/day genistein, or combination therapy, and were normalized to body weight (% body weight). The graphs present the results obtained for individual animals and mean ± standard deviation (SD). The means of untreated MPS I and wild-type (WT) control animals are depicted as red solid and green dashed horizontal lines, respectively. *N* = 6 animals per group. Significance: ** *p* < 0.01, *** *p* < 0.001, or **** *p* <0.0001 between treatments (Tukey’s post-hoc test after ANOVA); ^†^ *p* < 0.05, ^††^ *p* < 0.01, or ^†††^ *p* < 0.001 between treated and untreated MPS I mice (Student’s *t*-test).

**Figure 4 ijms-25-02371-f004:**
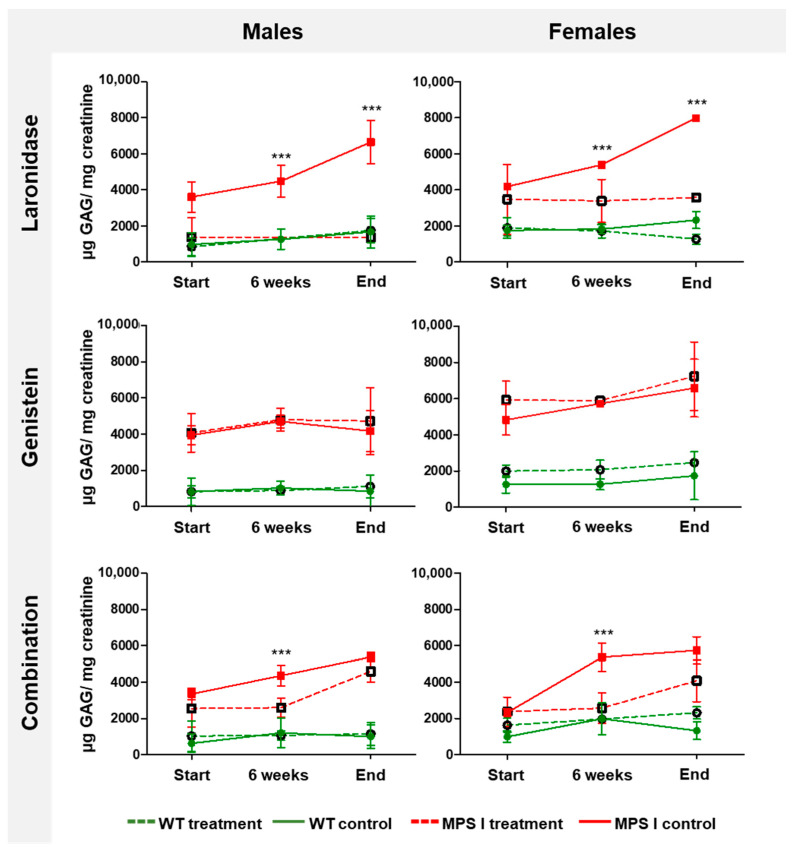
Total urinary GAG excretion in MPS I and wild-type (WT) mice of both sexes during treatment with laronidase, genistein, or combination therapy. GAG levels were measured in urine obtained before treatment (start) and following the administration of 100 U/kg/week of laronidase, 160 mg/kg/day of genistein, and combination therapy for 6 and 12 weeks (end). Total GAG levels were normalized to urinary creatinine levels. MPS I mice are marked in red, and WT mice, in green; dashed lines indicate treated, and solid lines indicate untreated controls. The results are presented as the mean ± standard deviation. *N* = 6 animals per group. Significance: *** *p* < 0.001 between matched treated and untreated MPS I or WT mice (Student’s *t*-test).

**Figure 5 ijms-25-02371-f005:**
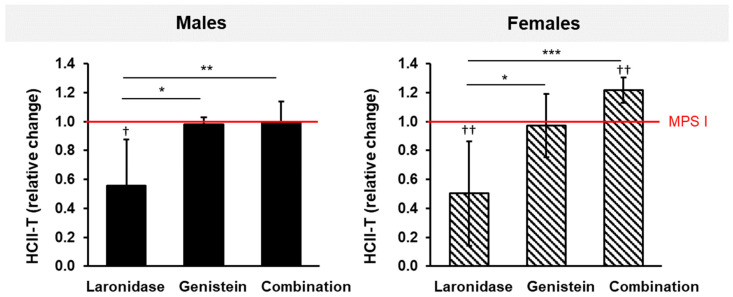
Relative changes in the HCII-T levels in the sera of MPS I mice of both sexes after treatment with laronidase, genistein, or combination therapy. HCII-T levels were measured in the sera obtained from mice after 12 weeks of treatment with 100 U/kg of laronidase, 160 mg/kg of genistein, or a combination of both, and were normalized to age-matched untreated MPS I animals (red horizontal, solid line). The results are presented as the mean ± standard deviation. *N* = 6 animals per group. Significance: * *p* < 0.05, ** *p* < 0.01, or *** *p* < 0.01, between treatments (Tukey’s post-hoc test after ANOVA or Dunn’s post-hoc test after Kruskal–Wallis test); ^†^ *p* < 0.05, ^††^ *p* < 0.01, between treated and untreated MPS I mice (Student’s *t*-test).

**Figure 6 ijms-25-02371-f006:**
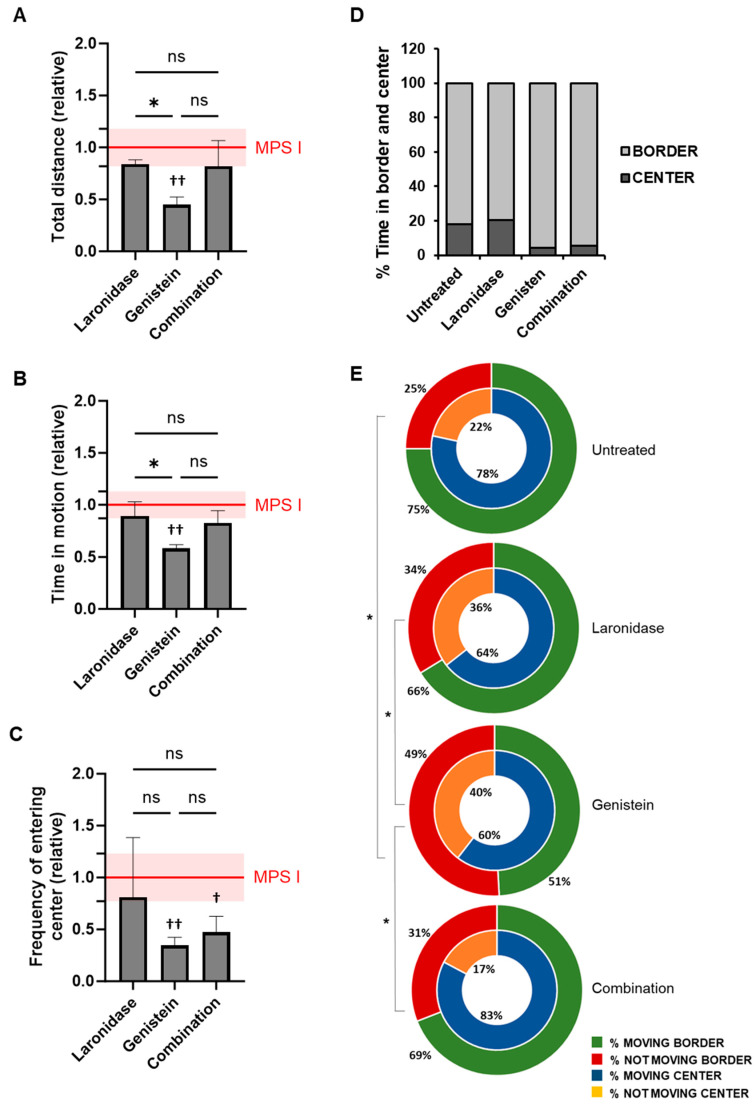
Behavior of six-month-old female MPS I mice after treatment with laronidase, genistein, or combination therapy. Behavior was assessed during a 30 min open field test for animals receiving 100 U/kg/week of laronidase, 160 mg/kg/day of genistein, or a combination of both for 12 weeks. The results of (**A**) the total distance covered during the test, (**B**) the total time that the animal remained in motion, and (**C**) the frequency of entering the center zone of the arena were normalized to those of untreated MPS I animals (red horizontal line, with the standard deviation indicated as a shaded field). The columns represent the mean ± standard deviation. *N* = 6 animals per group. Significance: * *p* < 0.05, between treatments (Tukey’s post-hoc test after ANOVA); ^†^ *p* < 0.05 or ^††^ *p* < 0.01 between treated and untreated MPS I mice (Student’s *t*-test); ns—not significant. (**D**) The percentage of time that mice spent in the center zone and the border of the arena, and (**E**) the percentage of time that animals were moving (blue or green) or not moving (orange or red), in the center of the arena (inner ring), or on its periphery (outer ring). The results are presented as mean percentages of time for all animals in each group. The untreated control group comprised age-matched female MPS I mice. *N* = 6 animals per group. Significance: * *p* < 0.05 (Tukey’s post-hoc test after ANOVA) between treatments regarding activity at the border of the arena. There were no statistically significant differences in the activity at the center of the arena for the same parameters between the groups.

## Data Availability

The data presented in this study are available upon request from the corresponding author.

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
