# Peer review of "Efficacy of a Combination Therapy with Laronidase and Genistein in Treating Mucopolysaccharidosis Type I in a Mouse Model"

_ijms, 2024, doi:10.3390/ijms25042371_

Round 1

Reviewer 1 Report

Comments and Suggestions for Authors

The authors reported interesting differences between the effects of therapies for mucopolysaccharidosis type I in female and male model animals and compared combined therapy with recombinant human α-L-iduronidase (laronidase)  and Genistein with monotherapies with either drug.

The title of the paper has been correctly attributed:

“Efficacy of combination therapy with laronidase and genistein 2 in treating mucopolysaccharidosis type I in a mouse model” because the author are assessing whether combined therapy is useful or not. I understand that combined therapy is to be compared with monotherapies with the same drugs.

This type of work is beneficial because it highlights the limits of enzyme replacement therapy in diseases with brain involvement and the importance of carefully stratifying model animals in terms of sex and age. The authors should not be afraid to present "negative" results; there are no "negative" results in this field as long as the experiments are conducted correctly. I believe the paper can be accepted after extensive review concerning, in particular, the possible benefits of combined therapy compared to monotherapies.

Figure 2 is crucial. We see that the combined therapy would have significant effects only in males' heart muscle and females' heart valves. Incomprehensibly, the effect in the brain of females is lower than that of monotherapies.   To make this more precise, the authors should add panels in which they compare combined therapy's effect with monotherapies and calculate the p-value for these, besides showing the comparison with non-treated controls. The same applies to Figure 3; the significance of joint therapy compared to monotherapies should be highlighted and calculated.

I don't understand Figure 4. I imagine that by “control” the authors refer to untreated sick (red) or healthy (green) animals. Why are  MPS values ​​different in the three panels? Why are wt-controls? Are these different animals? I believe the control should be the average of all untreated animals, two for MPS I animals, males and females, and two for WT animals, males and females.   Also, in Figure 7, the significance of the difference in the results obtained with the combined treatment and the monotherapies should be calculated, as well as showing the comparison with non-treated controls.   In light of a direct comparison between combined therapy and monotherapies, it could emerge that the statement “In this study, we demonstrated that a combination of laronidase at a dose of  100 U/kg/week and genistein at a dose of 160 mg/kg /day is an effective treatment for MPS I.” is overestimated.   Apart from this, the authors should recognize that their work is based on knockout mice and that in the case of missense mutations, different results could be obtained, and different forms of combination therapy could be used.

Furthermore, it should be mentioned that in cases of missense mutations, it is possible to propose pharmacological chaperones. These were developed for mucopolysaccharidosis type 1, (please cite Lin, Hung-Yi, et al. "Discovery of small-molecule protein stabilizers toward exogenous alpha-l-iduronidase to reduce the accumulated heparan sulfate in mucopolysaccharidosis type I cells." European Journal of Medicinal Chemistry 247 (2023): 115005.) and that pharmacological chaperones and/or reversible inhibitors of Alpha-L-iduronidase can be used for combined therapy as demonstrated for other lysosomal storage diseases ( please cite Iacobucci, Ilaria, et al. "Enzyme Replacement Therapy for FABRY Disease: Possible Strategies to Improve Its Efficacy." International Journal of Molecular Sciences 24.5 (2023): 4548.)

Reviewer 2 Report

Comments and Suggestions for Authors

Although the argument is quite interesting, the authors, in my opinion, misinterpreted the results and the final conclusion are not in line with the data presented. The data should be reported more clearly along with the statistical analysis description ( just to make an example: specify what the asterisks refer to, include wt in the graph etc).

The authors analyzed the effects of a monotherapy with laronidase or genistein and a combination therapy in a KO mouse model of MPSI . They evaluated and compared the effects of these three therapy on GAG storage in different tissues, the effect on the hepatosplenomegaly, urinary GAG excretion, locomotion and anxiety. They conclude that a combination therapy is is an effective treatment for MPS I. They also say that during long-term administration of high-dose genistein, some of this compound may have entered the central nervous system where, by decreasing GAG synthesis, it slowed neurodegeneration .

The work is interesting and all the studies that can improve the pharmacology of mucopolysaccharidosis are welcome. I appreciate the idea that the author have to combine different therapies and also the test done seems appropriate to me. Despite this I have doubts about this work. I hope that my comments and suggestion could help.

Major points

Results:

I recommend to include in all the captions more information about the statistical analysis: specify what you are comparing (MPSI treated vs WT treated or MPSI untreated) and specify the statistical test used. Moreover the WT treated data should be included in the paper and statistical analysis should also include the comparison between MPSI treated and untreated with the WT (treated and untreated). Once these analyzes have been done the results can be critically read and discussed. Please make the figures/graphs more clear, more easy to read (for example insert a legend with the color line etc and write some information about the test, the treatment or the sex in the title of the graph. The graph should also show WT first (on the left of the graph, fist column) then the model, then the treatment. WT treatment can be shown in the supplementary file.

1 The combination of laronidase and genistein reduces tissue GAG storage in MPS I mice

The authors write: "Genistein was most effective in decreasing GAG storage in the liver of MPS I mice of both sexes and in the spleen of male MPS I mice”. I disagree with this comment on the results. Observing the graphs the enzyme is the most effective in reducing the GAG, at least taking into consideration figure 2. Authors should compare the GAG of MPSI treated animals with the control (WT treated and untreated). After that I think that we can get a more critical reading of the data.

More, the combination therapy has an effect comparable to that of the enzyme alone except for heart valves of females mice.

2. Combination of laronidase and genistein reduces hepatosplenomegaly in MPS I mice

The authors should specify that the liver is not statistically different between MPS1 and WT if we consider males (I can’t see asterisk, so I imagine that MPS I (liver weigh) vs wt is not statistically different).

Moreover the authors say that the combination therapy resulted in a significant decrease in the liver weight of both male and female MPS I mice, this is true but this decrease depends on the enzyme therapy, not genistein, in fact Genistein itself did not affect the organ weight - This should be specify in the results section.

3 The combination of laronidase and genistein reduces urinary GAG excretion in MPS I mice

Same suggestion for this paragraph. Only the enzyme has an effect , the combination therapy has affect only because of the presence of laronidase. All the results are in line with the loss of effects on combination therapy on HCII-T level.

4 Effect of laronidase, genistein, and combination therapy on locomotor activity and anxiety behavior of MPS I mice

The open field test is used to assay general locomotor activity levels and anxiety. In Figure 6, MPSI female are less anxious, that is in contrast with other mouse model of MPSI in which anxiety index is increased. Then the author use genistein and observed a decreased of the frequency and time of explorations, that sounds wired.

The increase of the frequency and time of explorations toward the white compartment is typical of the diazepam treatment or genistein , drugs that are able to induce anxiolytic effect. In this work genistein have an opposite effect. Can you show the WT animals after treatment? This could help.

Discussion:

The discussion need to be improved. The authors say that “genistein reduced GAG levels in the brains of MPS I females and normalized anxiety and cognitive behaviors. The animals became calmer, moved less chaotically, and exhibited an increased sense of fear. This suggests that during long-term administration of high-dose genistein, some of this compound may have entered the central nervous system where, by decreasing GAG synthesis, it slowed neurodegeneration”. It is known that genistein affect the serotonergic system and this is the mechanism link to the anxiolytic-like effect. The authors cannot conclude that the behavioral change can be due to the “minimal” effect of GAG of the SNC. This should be included in the discussion as the most probable mechanism.

Round 2

Reviewer 1 Report

Comments and Suggestions for Authors

the paper is now suitable for publication

Reviewer 2 Report

Comments and Suggestions for Authors

The manuscript is signofonctly improuved. This revised version is ready for pubblication.